# Evaluation of Phenotypic and Genotypic Variations of Drug Metabolising Enzymes and Transporters in Chronic Pain Patients Facing Adverse Drug Reactions or Non-Response to Analgesics: A Retrospective Study

**DOI:** 10.3390/jpm10040198

**Published:** 2020-10-27

**Authors:** Victoria Rollason, Célia Lloret-Linares, Kuntheavy Ing Lorenzini, Youssef Daali, Marianne Gex-Fabry, Valérie Piguet, Marie Besson, Caroline Samer, Jules Desmeules

**Affiliations:** 1Division of Clinical Pharmacology and Toxicology, Department of Anesthesiology, Pharmacology, Emergency Medicine and Intensive Care, Geneva University Hospitals, 1205 Geneva, Switzerland; kuntheavy-roseline.ing@hcuge.ch (K.I.L.); youssef.daali@hcuge.ch (Y.D.); valerie.piguet@hcuge.ch (V.P.); marie.besson@hcuge.ch (M.B.); caroline.samer@hcuge.ch (C.S.); jules.desmeules@hcuge.ch (J.D.); 2Faculty of Medicine, Geneva University, 1206 Geneva, Switzerland; 3Ramsay Générale de Santé, Hôpital Privé Pays de Savoie, Maladies Nutritionnelles et Métaboliques, 74000 Annemasse, France; celialloret@yahoo.fr; 4Division of Psychiatric Specialties, Department of Psychiatry and Mental Health, Geneva University Hospitals, 1226 Thônex, Switzerland; marianne.gex-fabry@unige.ch

**Keywords:** personalised medicine, cytochrome P450, P-glycoprotein, COMT, analgesic drugs, adverse drug reaction

## Abstract

This retrospective study evaluates the link between an adverse drug reaction (ADR) or a non-response to treatment and cytochromes P450 (CYP), P-glycoprotein (P-gp) or catechol-O-methyltransferase (COMT) activity in patients taking analgesic drugs for chronic pain. Patients referred to a pain center for an ADR or a non-response to an analgesic drug between January 2005 and November 2014 were included. The genotype and/or phenotype was obtained for assessment of the CYPs, P-gp or COMT activities. The relation between the event and the result of the genotype and/or phenotype was evaluated using a semi-quantitative scale. Our analysis included 243 individual genotypic and/or phenotypic explorations. Genotypes/phenotypes were mainly assessed because of an ADR (*n* = 145, 59.7%), and the majority of clinical situations were observed with prodrug opioids (*n* = 148, 60.9%). The probability of a link between an ADR or a non-response and the genotypic/phenotypic status of the patient was evaluated as intermediate to high in 40% and 28.2% of all cases, respectively. The drugs in which the probability of an association was the strongest were the prodrug opioids, with an intermediate to high link in 45.6% of the cases for occurrence of ADRs and 36.0% of the cases for non-response. This study shows that the genotypic and phenotypic approach is useful to understand ADRs or therapeutic resistance to a usual therapeutic dosage, and can be part of the evaluation of chronic pain patients.

## 1. Introduction

The aim of any analgesic treatment is to improve the quality of life by decreasing pain while minimizing the potential toxicity of the treatment. Increasing knowledge on the pathophysiology of pain and the mechanisms of action of drugs have allowed strengthened guidelines based on the type of pain [1,2,3]. Despite well-accepted and largely used guidelines, a percentage of patients with pain are still under-treated or experience overwhelming adverse drug reactions (ADRs) at a usual therapeutic dosage, and this could be partly related to their genetic make-up.

Research suggests that exploring patients, mainly via a genotyping approach, for several metabolic pathways or drug targets may allow healthcare professionals to better explain patient response to analgesic treatment [4,5,6]. Despite this growing evidence, data are missing to justify the usefulness of a systematic assessment of the variability in drug metabolism in pain settings. The dosage and the choice of analgesics is usually based on the patient’s response and ADRs to a given treatment. Delaying the overall benefit of the therapeutic regimen may have negative effects on the quality of life and on the relationship between the patient and the healthcare professional. This also results in pain management becoming very time consuming. The economic burden of chronic pain is greater than most other health conditions due to its effects on the rates of absenteeism, reduced levels of productivity and increased risk of leaving the labor market [7,8,9].

At the Geneva University Hospitals, clinical pharmacologists actively contribute to the multidisciplinary pain center and answer physicians’ daily questions concerning ADRs, drug–drug interactions, non-response to treatment or therapeutic management (choice of the drug, dose and route of administration), among other situations. Patients are referred to the pain center for chronic refractory pain, resistance to opioid analgesics or unexpected ADRs in spite of usual therapeutic dosage. In order to clarify pharmacological abnormalities, genetic or phenotypic investigations may be proposed based on the clinical context, the nature of the ADR or the non-response and the concomitant medications.

In a previous retrospective study performed in a psychiatric setting, we demonstrated that variability in response to psychotropic drugs is related to a variation of the metabolic status, with an intermediate to high probability in one third of patients [10].

The present retrospective study aims to assess the extent that an ADR or a non-response to an analgesic treatment is related to a variation of cytochromes P450 (CYP), P-glycoprotein (P-gp) or catechol-O-methyltransferase (COMT) activity.

## 2. Materials and Methods

### 2.1. Patients and Setting

This study was a retrospective analysis of data collected during consultations with pain patients referred to our centre, using the same methodology as a previous analysis on patients in a psychiatry setting [10].

At the Geneva University Hospitals, the division of clinical pharmacology is the referral centre that coordinates the management of chronic refractory pain for all medical specialties. Questions on the therapeutic management of chronic pain come from hospital physicians and from private practices. In most cases, patients are seen during a consultation that allows understanding of the specific signs and symptoms of the disease, as well as the medication history. Individualised propositions are outlined in a report, and genetic and/or phenotypic investigations are considered when deemed appropriate by a senior clinical pharmacologist. Therefore, the potential genetic interactions for the drug reactions are decided ahead of time for each subject. If the tests are done, their results are then analysed according to the clinical context and summarised in a second report.

As described previously [10], we retrospectively collected results of the genetic and/or phenotypic investigations performed between January 2005 and November 2014, and selected all data related to analgesic drugs. Data were excluded if investigations did not assess one major metabolic pathway of the involved drug, according to the table of cytochrome substrates and as previously done for a study on psychotropic drugs only [10,11].

In several selected situations (*n* = 33), variants for the *COMT*, linked to central sensitization, were also explored.

The study was approved by the local ethics committee (Reference: 15-080) and was performed according to the Declaration of Helsinki and its later amendments or comparable ethical standards.

### 2.2. Evaluation Criteria

All clinical pharmacology reports were carefully reviewed and classified into different categories according to clinical events, namely ADR and non-response to the prescribed analgesics. Two experienced clinical pharmacologists, also pain specialists, independently assessed the possible association between these clinical events and the genetic and phenotypic results according to a semi-quantitative scale and to their clinical judgment, as previously described (Appendix A) [10]. The semi-quantitative scale was mainly built on scientific databases (e.g., Interactions médicamenteuses, cytochromes P450 et P-glycoprotéine (Pgp) [11]; DrugBank [12]; the Pharmacogene Variation (PharmVar) Consortium database [13]). For a given drug, each relevant major or minor metabolic pathway was considered, and a global rating was made according to a three-point scale: 0 = no or low probability of genetic and phenotypic results being linked with a clinical or biological problem; 1 = intermediate probability; 2 = high probability [10], meaning that a single score was given for each occurrence of a drug, an event (ADR or inefficacy), and all metabolic pathways of the specific drug. For drugs with active metabolites, the use of the table was completed by the available literature on the respective clinical relevance of the metabolite and parent compound. In case of disagreement, the opinion of a third expert in clinical pharmacology was sought and retained as the final score. The patient’s treatment at the time of the genetic and/or phenotypic investigation was recorded and taken into account when rating the association between the metabolic status and event.

### 2.3. Explored Pathways and Metabolic Status

Different enzymes and proteins were investigated: the phase I enzymes CYPs, the phase II enzyme COMT and the transmembrane transporter P-gp (encoded by the *ABCB1* gene).

Specific implication of the different CYPs and P-gp for the analgesics can be found in Table 1. Regarding *COMT*, studies have shown that polymorphisms have an impact on opioid dose, with the need for higher opioid equivalent dose in the mutated genotypes compared to the wild-type [14].

Activities of CYP2D6, 2C9 and 2C19 were assessed by phenotyping and genotyping. Activities of CYP1A2 and 3A4 were only determined by phenotyping, and *ABCB1* (encoding P-gp) and *COMT* polymorphisms were only assessed by genotyping.

#### 2.3.1. Genotyping

Until 2007, CYP2D6 genotyping was performed by real-time polymerase chain reaction (PCR) to detect the following defective mutations of *CYP2D6*: **3, *4, *5, *6, *35, *41* and duplications. From 2007, the microarray technology was introduced, and genotyping was performed on the AmpliChip CYP450 test from Roche, allowing the simultaneous analysis of 33 CYP2D6 alleles. For the *CYP2C9* and *CYP2C19* genes, the following variants were genotyped: *CYP2C9 *2* and **3*, *CYP2C19 *2* and **17* (since 2009) as previously described [69]. *ABCB1 c.3435C > T* (rs1045642) and *c.2677G > T* (rs2032582) polymorphisms were determined in a single multiplex PCR, with fluorescent probe melting temperature analysis on a LightCycler (Roche, Rotkreuz, Switzerland) as previously described [70]. The *COMT* genotype was assessed by focusing on single-nucleotide polymorphisms (rs4680) using a commercially available TaqMan single-nucleotide polymorphism genotyping assay (Applied Biosystems, Warrington, United Kingdom), as previously described [71].

The predicted phenotypes of the genotypes were based on enzyme activities of these alleles, as listed in the Pharmacogene Variation (PharmVar) Consortium database [13], the PharmGKB database [72] or on the instructions of the AmpliChip CYP450 2D6 test. Patients were classified as a poor metaboliser (PM), intermediate metaboliser (IM), normal metaboliser (NM) and ultra-rapid metaboliser (UM) for CYP2D6 and CYP2C19, and as having an increased, normal or reduced activity for CYP2C9.

For *ABCB1* and *COMT*, the variants that were genotyped coded for a reduced activity.

#### 2.3.2. Phenotyping

As in our previous study [10], phenotyping consisted in the administration of probe substrates metabolised by specific CYPs, and the determination of plasma, blood or urine metabolic ratios. Probe substrates used in our study were those of the Geneva cocktail, with caffeine for CYP1A2, flurbiprofen for CYP2C9, omeprazole for CYP2C19, dextromethorphan for CYP2D6 and midazolam for CYP3A [73]. At the beginning of our phenotypic investigations, CYP2D6 phenotyping was performed by assessing the metabolic ratio between dextromethorphan (DEM) and its metabolite dextrorphan (DOR) in the urine collected eight hours after ingestion of a single 25 mg oral dose of DEM [74,75]. The development of dried blood spot dosages has led to this method being seldomly carried out. Simultaneous phenotyping corresponded to the concomitant administration of multiple specific probes, with caffeine 50 mg, flurbiprofen 10 mg, omeprazole 10 mg, dextromethorphan 10 mg and midazolam 1 mg. Capillary blood samples two hours following drug administration allowed for measuring the activity of multiple CYP enzymes simultaneously, as previously reported [73,76].

Phenotypic classification was based on plasma or urine metabolic ratios (different for each method) according to a validated method developed in the laboratory of clinical pharmacology and toxicology of the Geneva University Hospitals [73,77,78]. CYP1A2, 2C9 and 3A4 enzyme activities were categorised as increased, normal or reduced. Results for CYP2C19 and 2D6 allowed for classifying patients as a poor metaboliser (PM), intermediate metaboliser (IM), normal metaboliser (NM) or ultra-rapid metaboliser (UM).

#### 2.3.3. Statistical Analysis

Categorical and continuous variables were described using frequency tables (*n*, %) and median (range), respectively. Comparisons of proportions were performed using Fisher’s exact tests. Inter-rater reliability of the scoring system was assessed with the kappa coefficient. Statistics were computed using SPSS version 22 (IBM Corporation, Armonk, NY, USA). All tests were two-tailed, with the significance level at 0.05.

## 3. Results

### 3.1. Patients, ADR or Non-Response, and Involved Analgesic Drugs

Between January 2005 and November 2014, 243 distinct evaluations involved ADRs or non-response to an analgesic drug, as presented in Table 2. These assessments were performed in 155 patients (104 women and 51 men), with a mean age of 55 years old (range: 1–99), because of one (*n* = 93), two (*n* = 46) or more (*n* = 16) ADRs or non-response to analgesic or co-analgesic treatment.

The majority of clinical situations were observed with prodrug opioids, namely tramadol, codeine, oxycodone and dextromethorphan (*n* = 148, 60.9%), followed by opioids (*n* = 57, 23.5%), antidepressants (*n* = 22, 9.1%), nonsteroidal anti-inflammatory drugs (NSAIDs) (*n* = 15, 6.2%) and paracetamol (*n* = 1, 0.4%). Genotypic or phenotypic explorations were mainly performed because of an ADR (*n* = 145, 59.73%) followed by a non-response (*n* = 92, 37.9%) at a usual therapeutic dosage of the analgesic. In six cases, the genotype and/or phenotype evaluation was done for both an ADR and a non-response to treatment (Table 2).

### 3.2. Metabolic Status

Patients’ CYP activity, assessed by genotyping and/or phenotyping, is presented in Table 3. Regarding the genotypic approach, the most frequently investigated enzyme was CYP2D6 (*n* = 105 patients), followed by the ABCB1 C3435T allele (n = 56), the ABCB1 G2677T allele (*n* = 54), CYP2C19 (*n* = 37), COMT (*n* = 33) and CYP2C9 (*n* = 23). For the phenotypic approach, the most frequently investigated enzyme was also CYP2D6 (*n* = 73 patients), followed by CYP3A (*n* = 32), CYP2C9 and CYP2C19 (*n* = 27 for both) and CYP1A2 (*n* = 21).

### 3.3. Concordance Between Predicted and Measured CYP Activity

Among the 73 patients who underwent a CYP2D6 phenotypic evaluation, 40 were also investigated by a genetic approach. In 52.5% of the cases (21/40), concordance was observed between the phenotype predicted from the genotype and the measured phenotype. Discordant results were observed in 19 patients. None of the nine patients with a CYP2D6 UM profile according to phenotype were detected by the result of the genotype (four *1/*1 patients, two *1/*2, two *2/*2 and one *1/*6). This cannot be explained by a co-medication, since no known inducer of CYP2D6 has ever been documented. Eight patients with an NM genotype (three*1/*1 patients, two *1/*4, one *2/*5, one *4/*35 and one *1XN/*4) showed an IM or PM phenotype. Four were co-medicated with CYP2D6 inhibitors, and one patient took numerous CYP2D6 substrates and phytotherapy. One patient with a UM genotype (*1/*2XN) showed an IM phenotype that could be explained by his co-medications. Finally, one patient with a PM genotype (*4/*4) was phenotypically an NM, and again, this case cannot be explained by co-medications, because there is no known inducer of CYP2D6. Overall, discordant results could be explained in five of the 19 cases.

Eight patients underwent CYP2C9 assessment by phenotypic and genotypic approaches, with a 50.0% concordance. For the four discordant results, no co-medication in the patients’ treatment could explain the genotypic/phenotypic discrepancy.

For CYP2C19, five patients were investigated by the two approaches. The genotype/phenotype concordance was observed in 100.0% of the cases.

The details of the discordances for CYP2C19 and CYP2D6, together with the CYP inhibitors possibly explaining the discordances, are presented in Table 4.

### 3.4. Link Between Metabolic Status and ADR or Non-Response

Inter-rater reliability of the scoring system was excellent (kappa = 0.95), with discrepant results between the two experts in only 5 of the 243 cases. Four cases were related to an ADR with morphine (two patients), buprenorphine and oxycodone. The last case was a non-response case with oxycodone.

As illustrated in Figure 1 and Table 5, the probability of a link between an ADR or a non-response and the metabolic status was rated globally as intermediate to high in 35.4% of all cases (*n* = 243). For an ADR and a non-response separately, proportions were 40.0% and 28.2%, respectively. The same results for specific analgesic classes showed that a link between an ADR or a non-response and the metabolic status for opioids was globally rated as intermediate to high in 29.6% of patients. For occurrence of an ADR, the link was evaluated as being 37.3%.

For prodrug analgesics, the link between an ADR or a non-response and the metabolic status was rated globally as intermediate to high in 41.4% of all cases. This link was evaluated as being intermediate to high in 45.7% of cases for occurrence of an ADR and in 35.9% for non-response.

Regarding specific analgesics (Table 6), the global probability of a link between an ADR or a non-response and the metabolic status was rated as intermediate to high in 39% of patients taking tramadol, 33% of patients taking morphine, 56.5% of patients taking codeine and 29.1% of patients taking oxycodone. More specifically, the probability of a link between an ADR and the metabolic status was rated globally as intermediate to high in 38.8% of patients taking tramadol, 37.5% of patients taking morphine, 68.5% of patients taking codeine and 28.6% of patients taking oxycodone. For a non-response, the link with the metabolic status was 39.4% for tramadol, 38.5% for codeine and 29.4% for oxycodone (Table 6).

Table 5 and Table 6 further document the link between metabolic status and an ADR or a non-response, with results for other drug classes (Table 5) and specific analgesics (Table 6), and with examples of each situation.

When taking into account only the cases with a measured CYP2D6 phenotype, the probability of a link between an ADR or a non-response and the metabolic status was rated globally as intermediate to high in 35.9% of patients taking tramadol, 45.5% of patients taking codeine and 41.7% of patients taking oxycodone. The proportions of patients in each CYP2D6 phenotype group did not significantly differ according to type of demand (ADR vs. non-response) for tramadol, codeine or oxycodone (Fisher’s exact test, *p* = 0.50, 0.071 and 0.78 respectively) (Table 7).

## 4. Discussion

This retrospective analysis shows that ADRs or non-response to treatment could be explained by a variation in the metabolic status of the patient in 35.4% of the cases, with an intermediate to high probability according to our semi-quantitative scale.

The overall rate for CYP2D6 and CYP2C9 concordance between the phenotype predicted by the genotype and the measured phenotype was close to 50%. Several reasons can be put forward to explain this. First, the genetic approach with the AmpliChip CYP450 test has a low sensitivity for UM prediction of CYP2D6, in line with a previous study [74]. Indeed, in nine cases (22.5%), UM phenotypes were deemed as NMs when the genotype was assessed. The other reason for discordance is related to the patient’s co-medications. For CYP2D6, among the ten discordant cases, half of the patients received a drug known to inhibit CYP2D6 at the time of the phenotyping procedure. For CYP2C9, the four discordant cases could not be explained by co-medications. An obvious reason for these discordances could be the set of genes and SNPs explored in this retrospective study. Indeed, rare genetic variants were not included, and these could explain some of the discordances observed between the genotype and the phenotype, as highlighted in recent publications and guidelines [79,80,81]. The genetic tests used for the analysis of our patients’ genotypes would assign the patient a wild-type status while carrying a rare IM/PM/UM variant. Other environmental factors, such as diet, smoking or alcohol consumption, as well as diseases, may also have an impact on the activity of CYPs.

Modification of CYP activity was more often detected when the cause for exploration was an ADR (Figure 1, Table 5 and Table 6). This seems quite understandable, because an ADR is probably easier to detect than a non-response to treatment, especially for analgesic drugs for which the cornerstone of adequate treatment is gradually titrating the dose until reaching the desired analgesia, not systematically taking into account an eventual lack of efficacy. Moreover, numerous studies have shown the link between reduced or increased activity of CYPs in patients experiencing ADRs with analgesics and prodrug opioids in particular [82,83,84]. For prodrug opioids, studies demonstrate that the PK of these analgesics is modified in UMs for codeine, tramadol, hydrocodone and oxycodone. Regarding the PD modifications, data for codeine and tramadol also show a marked decrease in analgesia and ADRs. A guideline is even available for codeine with suggestions on how to adapt the treatment in the case of a modified phenotype [81]. There is no specific guideline for tramadol, but as tramadol is also a prodrug metabolized through CYP2D6, it seems reasonable to apply the same recommendations as those suggested for codeine, even though tramadol has a more complex mechanism of action [85]. For hydrocodone and oxycodone, further clinical research is still needed to be able to create specific guidelines [81,84,86].

Regarding non-efficacy, our study shows that for prodrug opioids, genotype and phenotype investigations help to give a better understanding of the reason for analgesic failure in 36% of cases. This link for analgesic failure was particularly high for tramadol and codeine, being 39.4% and 38.5%, respectively. Indeed, studies have clearly shown the association between the CYP2D6 phenotype of the patient and the PK and PD of the prodrug opioids, with a decreased analgesia and a decreased occurrence of ADRs in PMs. This has been clearly illustrated for codeine and tramadol. For hydrocodone and oxycodone, studies show a link with the PK profile of these two drugs, but clinical studies are contradictory and still fail to demonstrate a clear lack of analgesia in PM patients [81,84,85,86].

This study has several limitations. First, its retrospective approach put in evidence a lack of systematic collection of the information given by the patients. Whereas the co-medications taken by the patients were thoroughly collected, other anamnestic information was sometimes lacking, such as knowing if the patient was a smoker or not. Moreover, the patients in our consultation often come to us after several attempts of different analgesic treatments, and medical history is done on their retrospective reminiscence of ADRs. Moreover, for the non-response to treatment, lack of compliance was only anamnestic in most cases. In addition, the semi-quantitative scale applied to the cases was not a validated scale, and may not be sensitive enough to properly rate the complexity of drugs that have several metabolic pathways and/or active metabolites. Finally, the sample size and thus statistical power were limited when analysing the association of CYP2D6 phenotype and type of demand for specific drugs. Due to the genotyping and phenotyping methods available at the beginning of the study, not all pathways of the analgesics were explored for every case. One example is for *ABCB1*, for which only genotyping of two alleles was available at the beginning of the study, whereas we now have a full phenotyping cocktail that allows us to also evaluate the real activity of P-gp. This would have been especially useful for the opioids. Another example is that CYP2C8 was not tested by phenotype or genotype, but has implications in the metabolism of several NSAIDs, in particular diclofenac and ibuprofen [35,36,41,42].

## 5. Conclusions

This study highlighted the usefulness of genotyping and/or phenotyping patients when they experience either an ADR or a non-response when treated with an analgesic drug in a real-life setting. A link between the occurrence of an ADR and a modified activity of CYPs or transporters was found in 40% of cases, this link being nearly 30% for non-response. This study showed an even higher link for both situations when the drug was a prodrug opioid. This is in line with the existing evidence that shows a strong link with CYP2D6-dependant metabolism and clinical outcomes [81,87]. As put forward in several guidelines, this genetic and phenotypic information can help choose the right drug at the right dosage for a specific patient [87].

Both the genotypic and phenotypic approach, with the methods available today, are fast and effective. Our experience showed that they are complementary approaches because, whereas the genotype shows the inherited polymorphisms of the genetic make-up of the patient, the phenotype takes into account the environmental modifications in the metabolic capacity of the patient, as well as the impact of diseases (e.g., inflammation) and especially the co-medications in these real-life situations. Therefore, a multidisciplinary approach in which both genotype and phenotype are assessed and clinical pharmacologists and pain specialists work hand in hand will result in improved pain management of these patients. However, these genotypic and phenotypic evaluations rely on characterizing a specific gene for a specific drug or drug class. The future seems to be in new technologies, for example those that look at the whole genome or computational physiologically-based pharmacokinetic/pharmacodynamic approaches (PBPK/PD) [88,89].

This study should open the door to further similar prospective investigations with validated questionnaires and structured data collection. This would surely help to define the most relevant clinical situations in which these approaches would be helpful.

## Figures and Tables

**Figure 1 jpm-10-00198-f001:**
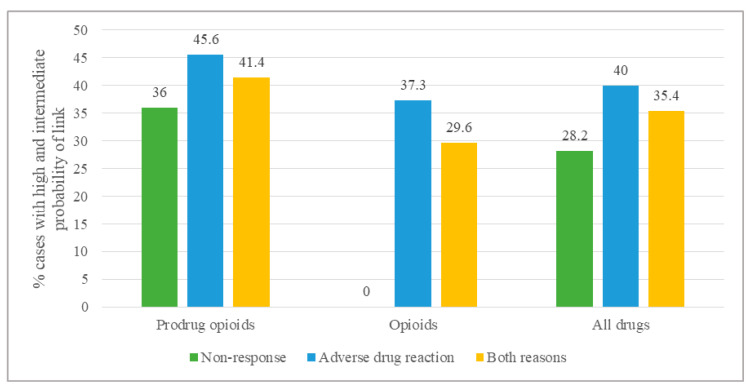
Prevalence of therapeutic problems attributable to an abnormal metabolic status, with an intermediate to high probability according to the analgesic class and the clinical event.

**Table 1 jpm-10-00198-t001:** Analgesics and their metabolic pathways [15,16,17,18,19,20,21,22,23,24,25,26,27,28,29,30,31,32,33,34,35,36,37,38,39,40,41,42,43,44,45,46,47,48,49,50,51,52,53,54,55,56,57,58,59,60,61,62,63,64,65,66,67,68].

	1A2	2C8	2C9	2C19	2D6	3A4/5	P-gp
amitriptyline					**!**	**!**	
buprenorphine							
celecoxib							
clomipramine				**!**		**!**	
codeine					**!**		
dextromethorphan					**!**		
diclofenac							
duloxetine							
fentanyl							
ibuprofen							
imipramine							
indomethacin							
ketoprofen							
mefenamic acid							
methadone							
morphine							
nortriptyline							
oxycodone					**!**		
tramadol					**!**		
trimipramine							
venlafaxine					**!**		

Major pathway 

; Minor pathway 

; Active metabolite 
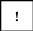
.

**Table 2 jpm-10-00198-t002:** Characteristics of patients (*n* = 155) and demands (*n* = 243).

	Frequency	%
Sex, female/male	104/51	67.1/32.9
Number of demands		
1	93	60.0
2	46	29.7
3 to 6	16	10.3
Genotype assessment ^a^	122	78.7
Phenotype assessment ^b^	78	50.3
Analgesic drug categories (*n* = 243)		
opioids ^c^	57	23.5
Prodrug opioids ^d^	148	60.9
Nonsteroidal anti-inflammatory drugs ^e^	15	6.2
antidepressants ^f^	22	9.1
paracetamol	1	0.4
Reasons for demands (*n* = 243)		
Adverse events	145	59.7
Non-response	92	37.9
Both	6	2.5

^a^ At least one among cytochrome P450 (CYP) 2D6, CYP2C9, CYP2C19, P-glycoprotein (P-gp) or catechol-O-methyltransferase (COMT). ^b^ At least one among CYP1A2, CYP2D6, CYP2C9, CYP2C19 or CYP3A. ^c^ Includes morphine, buprenorphine, fentanyl and methadone. ^d^ Includes tramadol, codeine, oxycodone and dextromethorphan. ^e^ Includes ibuprofen, mefenamic acid, diclofenac, indomethacin, ketoprofen and celecoxib. ^f^ Includes amitriptyline, clomipramine, trimipramine, imipramine, nortriptyline, duloxetine and venlafaxine.

**Table 3 jpm-10-00198-t003:** Genotype, phenotype predicted from genotype and measured phenotype (*n* = 155 patients).

	Genotype	Predicted Phenotype		Measured Phenotype	
		Frequency	%	Frequency	%
CYP2D6		*n* = 105		*n* = 73	
UM	*1/*2xN	3	2.9	15	20.6
NM	*1/*1, *1/*2, *1/*3, *1/*4, *1/*5, *1/*6, *1/*10, *1/*35, *1/*41, *1XN/*4, *2/*2, *2/*4, *2/*5, *2/*6, *2/*10, *2xN/*4, *4/*35, *10/*35, *35/*35, *35/*41	84	80.0	27	37.0
IM	*4/*9, *4/*41, *4/*10XN, *4XN/*41, *5/*41, *9/*41, *10/*41, *17/*41	11	10.5	19	26.0
PM	*4/*4, *4/*5	7	6.7	12	16.4
CYP2C9		*n* = 23		*n* = 27	
increased		0	0.0	5	18.5
normal	*1/*1	15	65.2	18	66.7
reduced	*1/*2, *1/*3	8	34.8	4	14.8
CYP2C19		*n* = 37		*n* = 27	
UM	*17/*17	1	2.7	2	7.4
NM	*1/*1	25	67.6	21	77.8
IM	*1/*2, *2/*17	10	27.0	2	7.4
PM	*2/*2	1	2.7	2	7.4
CYP1A2				*n* = 21	
increased				10	47.6
normal				11	52.4
reduced				0	0.0
CYP3A				*n* = 32	
increased				4	12.5
normal				24	75.0
reduced				4	12.5
ABCB1 C3435T (rs1045642)		*n* = 56			
normal	CC	13	23.2		
reduced	CT	30	53.6		
reduced	TT	13	23.2		
ABCB1 G2677T (rs2032582)		*n* = 54			
normal	GG	18	33.3		
reduced	GT/GA	28	51.9		
reduced	TT	8	14.8		
COMT rs4680		*n* = 33			
normal	GG	4	12.1		
reduced	GA	21	63.6		
reduced	AA	8	24.2		

UM: ultra-rapid metaboliser, NM: normal metaboliser, IM: intermediate metaboliser, PM: poor metaboliser, CYP: cytochrome P450 COMT: catechol-O-methyltransferase.

**Table 4 jpm-10-00198-t004:** Concordance between phenotype predicted from genotype and measured phenotype.

Predicted Phenotype	*n*	% Concordance ^a^	Measured Phenotype				Co-medication Possibly Relevant to Discordant Cases (Predicted > Measured Phenotype)
			UM Frequency	NM Frequency	IM Frequency	PM Frequency	
CYP2D6	40	52.5					
UM			0	0	1	0	citalopram (UM > IM)
NM			9	14	7	1	fluoxetine (NM > IM); citalopram (NM > IM); duloxetine (NM > IM); multiple CYP2D6 substrates and phytotherapy (NM > IM)
IM			0	0	6	0	
PM			0	1	0	1	
CYP2C9	8	50.0					
NM			2	3	0	0	
IM			0	2	1	0	
CYP2C19	5	100.0					
NM			0	5	0	0	

^a^ Percent concordance of measured phenotype with phenotype predicted from genotype. CYP: cytochrome P450, UM: ultra-rapid metaboliser, NM: normal metaboliser, IM: intermediate metaboliser, PM: poor metaboliser.

**Table 5 jpm-10-00198-t005:** Linkage between the type of demand and patient genotype or phenotype by analgesic group.

Type of Demand	Linkage with Genotype or Phenotype			Examples
	Estimated Probability ^a^	Frequency	%	
Adverse events				
All drugs (*n* = 145)	intermediate	30	20.7	-
	high	28	19.3	-
Opioids (*n* = 43)	intermediate	14	32.6	Marked drowsiness for 24 h with buprenorphine; homozygous mutation for ABCB1 C3435T: TT and heterozygous mutation for G2677T/A: GT
	high	2	4.7	Prolonged altered consciousness state with fentanyl; CY3A4 PM phenotype
Prodrug opioids (*n* = 81)	intermediate	13	16.0	Malaise with dextromethorphan; CYP2D6 IM genotype and phenotype
	high	24	29.6	Severe vomiting with codeine; CYP2D6 UM phenotype
NSAIDs (*n* = 5)	intermediate	2	40.0	Numerous adverse reactions with diclofenac; CYP2C9 IM genotype
	high	0	0.0	-
Antidepressants (*n* = 16)	intermediate	1	6.3	Numerous adverse reactions with imipramine; CYP2D6 IM genotype and phenotype
	high	2	12.5	Drowsiness and confusion with trimipramine; CYP2D6 PM phenotype
Non-response				
All drugs (*n* = 92)	intermediate	12	13.0	-
	high	14	15.2	-
Opioids (*n* = 11)	intermediate	0	0.0	-
	high	0	0.0	-
Prodrug opioids (*n* = 64)	intermediate	12	18.8	Non-response to oxycodone
	high	11	17.2	Non-response to codeine; CYP2D6 PM phenotype
NSAIDs (*n* = 10)	intermediate	0	0.0	-
	high	1	10.0	Non-response to ibuprofen; CYP2C9 UM phenotype
Antidepressants (*n* = 6)	intermediate	0	0.0	-
	high	1	16.7	Non-response to amitriptyline; CYP2D6 PM phenotype

^a^ Consensus among two to three raters with experience in clinical pharmacology. UM: ultra-rapid metaboliser, IM: intermediate metaboliser, PM: poor metaboliser, CYP: cytochrome P450.

**Table 6 jpm-10-00198-t006:** Link between the type of demand and patient genotype or phenotype by specific drug.

Type of Demand	Linkage with Genotype or Phenotype			Examples
	Estimated Probability ^a^	Frequency	%	
Adverse events				
All drugs (*n* = 145)	intermediate	30	20.7	-
	high	28	19.3	-
Tramadol (*n* = 49)	intermediate	7	14.3	Sedation, dizziness; CYP2D6 IM genotype and phenotype
	high	12	24.5	Sedation and hallucinations; CYP2D6 UM genotype
Morphine (*n* = 32)	intermediate	12	37.5	Comateous state; homozygous mutation for COMT rs 4680 A/A
	high	0	0.0	-
Codeine (*n* = 19)	intermediate	4	21.1	Hallucination; CYP2D6 IM genotype
	high	9	47.4	Important drowsiness; CYP2D6 UM phenotype
Oxycodone (*n* = 7)	intermediate	1	14.3	Drowsiness; CYP2D6 IM genotype
	high	1	14.3	Disorientation and sedation; CYP2D6 UM phenotype
Non-response				
All drugs (*n* = 92)	intermediate	12	13.0	-
	high	14	15.2	-
Tramadol (*n* = 33)	intermediate	7	21.2	Non-response; CYP2D6 IM phenotype
	high	6	18.2	Non-response; CYP2D6 PM phenotype
Morphine (*n* = 4)	intermediate	0	0.0	-
	high	0	0.0	-
Codeine (*n* = 13)	intermediate	4	30.8	Non-response; CYP2D6 IM phenotype
	high	1	7.7	Non-response; CYP2D6 PM phenotype
Oxycodone (*n* = 17)	intermediate	1	5.9	Non-response; CYP2D6 IM phenotype
	high	4	23.5	Non-response; CYP2D6 PM phenotype

^a^ Consensus among two to three raters with experience in clinical pharmacology. UM: ultra-rapid metaboliser, IM: intermediate metaboliser, PM: poor metaboliser, CYP: cytochrome P450, COMT: catechol-O-methyltransferase.

**Table 7 jpm-10-00198-t007:** Association between the type of demand and measured phenotype.

Drug	Type of Demand	*n*	Measured CYP2D6 Phenotype				*p*-Value ^a^
			UM Frequency (%)	NM Frequency (%)	IM Frequency (%)	PM Frequency (%)	
Tramadol							0.50
	Adverse events	24	6 (25.0)	9 (37.5)	6 (25.0)	3 (12.5)	
	Non-response	15	1 (6.7)	6 (40.0)	6 (40.0)	2 (13.3)	
Codeine							0.071
	Adverse events	13	5 (38.5)	6 (46.2)	1 (7.7)	1 (7.7)	
	Non-response	9	0 (0.0)	4 (44.4)	4 (44.4)	1 (11.1)	
Oxycodone							0.78
	Adverse events	3	1 (33.3)	2 (66.7)	0 (0.0)	0 (0.0)	
	Non-response	9	3 (33.3)	2 (22.2)	1 (11.1)	3 (33.3)	

^a^ Fisher’s exact test for equal proportions of CYP2D6 phenotypes according to type of demand. CYP: cytochrome P450, UM: ultra-rapid metaboliser, NM: normal metaboliser, IM: intermediate metaboliser, PM: poor metaboliser.

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
