# Peer review of "Evaluation of Phenotypic and Genotypic Variations of Drug Metabolising Enzymes and Transporters in Chronic Pain Patients Facing Adverse Drug Reactions or Non-Response to Analgesics: A Retrospective Study"

_jpm, 2020, doi:10.3390/jpm10040198_

Round 1
Reviewer 1 Report
Dear Authors,
Thank you for provising me with the oppurtunity to review your article. Generally, the quality of prsentation of the article is fine and the following comments can help you to improve its quality of presentation:
Please start the methods section with the recognition of the research design, in connection to the previously published research.
Ethical considerations applying to the research process should be fully described under a separate subheading.
Please make the figure colourful.
Good luck!
Author Response
Dear Reviewer,
Thank you for your suggestions that have been very helpful. Please find below a point-by point answer:
Please start the methods section with the recognition of the research design, in connection to the previously published research.
Line 75-77. The paragraph "2.1. Patients and setting" was reorganised to mention the design of the study, with reference to the previously published study at the beginning of the paragraph.
Ethical considerations applying to the research process should be fully described under a separate subheading
Line 96-97. Ethical considerations were moved to a separate paragraph at the end of "2.1. Patients and setting" subheading
Please make the figure colourful
Line 244. This was done, we hope it looks better
Thank you again and kind regards
Reviewer 2 Report
Thw work is very interesting and well defined.I have only two minor requests
1) Improve the quality of Figure 1.
2) Improve the future perspetives in conclusions to underline the importance of obtained results.
Author Response
Dear Reviewer,
Thank you very much for your helpful suggestions. Please find below a point-by-point answer to your comments:
Improve the quality of Figure 1
Line 244. This was done, we hope it looks better
Improve the future perspectives in conclusions to underline the importance of obtained results
Line 336-358. The conclusion and perspectives were improved
Thank you again and kind regards
Reviewer 3 Report
This is a retrospective study in personalized medicine field and it reveals pharmacogenetic impact in adverse drug response and pharmacological resistance in patients taking analgesic drugs for chronic pain. In my opinion this study is of interest, clearly organized and well described, with novelty aspects. I think there is a bit of sorting out the description of enzymes you have investigated and it would be worth to indicate phase I, phase II enzymes and transporters. Additionally, it would be interested to shortly describe the involvement of studied enzymes in the metabolizing of given analgesic drug(s).
Author Response
Dear Reviewer,
Thank you very much for your helpful suggestions. Please find below a point-by-point answer to your comments:
This is a retrospective study in personalized medicine field and it reveals pharmacogenetic impact in adverse drug response and pharmacological resistance in patients taking analgesic drugs for chronic pain. In my opinion this study is of interest, clearly organized and well described, with novelty aspects. I think there is a bit of sorting out the description of enzymes you have investigated and it would be worth to indicate phase I, phase II enzymes and transporters.
Line 119-123. We clarified this in the section 2.3 by adding a paragraph better explaining the enzymes.
Additionally, it would be interested to shortly describe the involvement of studied enzymes in the metabolizing of given analgesic drug(s).
Line 121-126. We added a table describing the metabolic pathways of the analgesics regarding CYPs and a text explaining the COMT and P-gp.
Thank you again and kind regards